# Changes in Parents’ COVID-19 Vaccine Hesitancy for Children Aged 3–17 Years before and after the Rollout of the National Childhood COVID-19 Vaccination Program in China: Repeated Cross-Sectional Surveys

**DOI:** 10.3390/vaccines10091478

**Published:** 2022-09-06

**Authors:** Xiaofeng Zhou, Shiyu Wang, Kechun Zhang, Siyu Chen, Paul Shing-fong Chan, Yuan Fang, He Cao, Hongbiao Chen, Tian Hu, Yaqi Chen, Zixin Wang

**Affiliations:** 1Longhua District Center for Disease Control and Prevention, Shenzhen 518110, China; 2Jockey Club School of Public Health and Primary Care, Faculty of Medicine, The Chinese University of Hong Kong, Hong Kong 999077, China; 3Department of Health and Physical Education, The Education University of Hong Kong, Hong Kong 999088, China

**Keywords:** COVID-19 vaccine hesitancy for children, changes, parents, repeated cross-sectional surveys, China

## Abstract

China started to implement COVID-19 vaccination programs for children in July 2021. This study investigated the changes in parents’ COVID-19 vaccine hesitancy for children before and after the vaccination program rollout. Repeated cross-sectional online surveys among full-time adult factory workers were conducted in Shenzhen, China. This analysis was based on 844 (first round) and 1213 parents (second round) who had at least one child aged 3–17 years. The prevalence of vaccine hesitancy for children aged 3–11 years dropped from 25.9% (first round) to 17.4% (second round), while such a prevalence for children aged 12–17 years dropped from 26.0% (first round) to 3.5% (second round) (*p* < 0.001). Positive attitudes, a perceived subjective norm, and perceived behavioral control related to children’s COVID-19 vaccination were associated with lower vaccine hesitancy in both rounds. In the second round and among parents with children aged 3–11 years, negative attitudes and exposure to information on SARS-CoV-2 infection after receiving a primary vaccine series were associated with higher vaccine hesitancy, while exposure to experiences shared by vaccine recipients and infectiousness of variants of concern were associated with lower vaccine hesitancy. Regular monitoring of vaccine hesitancy and its associated factors among parents should be conducted to guide health promotion.

## 1. Introduction

Worldwide, coronavirus disease 2019 (COVID-19) is a serious public health threat that has affected the health of children [1,2]. Children under the age of 18 years accounted for 17.5% of all COVID-19 cases in the United States [3], 16.7% in Germany [4], 11.7% in India [5], and 2% in France [6]. In Hong Kong, China, children represented roughly 6.2% of all cases during the fifth wave of the COVID-19 outbreak [7]. As compared with adults, children have a lower death rate and lower number of hospitalizations and intensive care unit admissions following SARS-CoV-2 infection [3,8]. However, considering their large population size (32% globally), COVID-19 vaccination for children is essential for pandemic control.

Systematic reviews and meta-analyses consistently demonstrated that the COVID-19 vaccination was highly effective in preventing COVID-19 among children aged 3–11 years and those aged 12–17 years, with a vaccine efficacy above 90% [9,10,11,12,13,14,15,16]. Studies also revealed favorable safety profiles of COVID-19 vaccination in children, with most side effects being mild to moderate (e.g., pain at the injection site, fever, headache) [10,12]. Serious side effects, such as myocarditis, were rare (70.7–105.9 per million doses of the vaccine in children aged 12–17 years) [17]. Therefore, the World Health Organization (WHO) recommends that children aged five years or above should receive the COVID-19 vaccination [18]. The WHO highlights that BNT162b2 can be safely administered to children aged ≥5 years and mRNA1273 can be used in children aged ≥12 years [18]. China approved the emergency use of two COVID-19 inactivated vaccines (Sinopharm and SinoVac-CoronaVac) in children aged 3–17 years on 11 June 2021 [19]. China started implementing COVID-19 vaccination for children aged 12–17 years in July 2021 and children aged 3–11 years in October 2021 [20,21].

The coverage of COVID-19 vaccination among children varied across countries. In Singapore, 80% and 98% of children aged 5–11 years and 12–19 years, respectively, completed the primary vaccination series in June 2022 [22]. In the United States, 29% of children aged 5–11 years and 59% of those aged 12–17 years completed the primary vaccine series [23]. A much lower uptake rate was found in the United Kingdom (7% and 24% of children aged 5–11 and 12–15 years, respectively) in May 2022 [24]. Globally, vaccine hesitancy is a significant challenge for COVID-19 vaccination programs and pandemic control. Parents are usually the decision makers for children aged 3–17 years or have a strong influence regarding their vaccination. It is hence important to investigate parents’ COVID-19 vaccine hesitancy for children and its associated factors. Meta-analyses reported an overall prevalence of COVID-19 vaccine hesitancy of 40% among parents [25,26]. Among parents, older age, having access to scientific information and recommendation, perceived higher threat of COVID-19, positive attitudes toward COVID-19 vaccines and other vaccination for themselves and their children, and parents’ uptake of COVID-19 vaccination were associated with lower COVID-19 vaccine hesitancy for children [25,26].

In China, at least 14 studies investigated parents’ or guardians’ COVID-19 vaccine hesitancy for their children; the majority (*n* = 10) were conducted before the rollout of the COVID-19 vaccination program for children [27,28,29,30,31,32,33,34,35,36,37,38,39,40]. In addition to factors identified by the aforementioned meta-analyses, education, income, parents’ psychological distress, and parents’ exposure to COVID-19 vaccine-related information on social media were also associated with vaccine hesitancy for children among Chinese parents [27,28,29,30,31,32,33,34,35,36,37,38,39,40]. During the pandemic, the unstable nature of the pandemic, rapid changes in vaccine-related policies, availability of vaccines, widespread reports about vaccines from the media, and the emergence of new variants of concern influenced perceptions and vaccine hesitancy for children [41,42,43]. Previous studies showed that the prevalence of and factors associated with vaccine hesitancy changed substantially before and after the rollout of mass COVID-19 vaccination programs [44,45,46]. To our knowledge, there was a dearth of studies that monitored the changes in parents’ COVID-19 vaccine hesitancy and associated factors at different stages before and after the rollout of childhood COVID-19 vaccination programs.

To address these knowledge gaps, we conducted two rounds of cross-sectional online surveys among Chinese parents. In China, the rollout of the vaccination program for children aged 3–11 years started later than that for children aged 12–17 years. It was possible that the level of vaccine hesitancy and associated factors would differ between parents of children aged 3–11 years and children aged 12–17 years. Therefore, this study aimed to investigate the changes in COVID-19 vaccine hesitancy before and after the rollout of COVID-19 vaccination programs for children in two groups of parents (i.e., parents of children aged 3–11 years and parents of children aged 12–17 years). We also compared the difference in COVID-19 vaccine hesitancy between these two groups of parents in the same round of the survey. We hypothesized that the prevalence of parents’ vaccine hesitancy would decrease after the rollout of national children’s vaccination programs, and factors associated with parents’ vaccine hesitancy would be different at different time points.

## 2. Materials and Methods

### 2.1. Study Design

The original study had two rounds of cross-sectional online surveys that looked at COVID-19 vaccination uptake and attitudes among adult factory workers in Shenzhen, China [40,47,48]. The first round of the survey was conducted from 1 to 7 September 2020, and the second round was implemented between 26 and 31 October 2021. The study sites, sampling, and data collection were identical between the two rounds of surveys [40,47,48]. This manuscript was based on a sub-sample of these participants who had at least one child aged 3–17 years. Shenzhen, bordering Hong Kong to the north, is a major special economic zone in China. The majority of the factories here are located in the Longhua district of Shenzhen. There were 1517 factories and more than one million factory workers in 2020 [49]. Shenzhen started to implement COVID-19 vaccination for children aged 12 to 17 years in July 2021 and for children aged 3 to 11 years in October 2021 [20,21]. We present the situation of COVID-19 and policy changes related to COVID-19 vaccination in Shenzhen in Figure 1.

### 2.2. Participants and Data Collection

Participants of these two rounds of online surveys were full-time employees of factories in Shenzhen and aged 18 years or above. In Shenzhen, all factory employees are required to complete a physical examination once a year at designated study sites. We selected the same study sites for recruitment in both surveys, which covered all six designated sites providing physical examinations to factory workers in the Longhua district. These sites included three public hospitals, two private hospitals, and the district’s Center for Disease Control and Prevention (CDC). During the recruitment period, CDC staff approached all adults attending these sites for physical examination, informed them about the study details, confirmed their eligibility, and invited them to complete an online survey on site. The CDC staff guaranteed that participation was voluntary, refusal would have no consequences, the survey would not collect personal contact information or identification, and data would be kept strictly confidential.

We used Questionnaire Star, which is an encrypted online survey platform commonly used in China, to carry out the surveys. At the testing sites, CDC staff invited prospective participants to scan a quick response (QR) code to access an electronic consent form and the online questionnaire. Participants signed the electronic consent form before they could fill out the online survey. The Questionnaire Star tool only allowed each mobile device to access the online questionnaire once to avoid duplication. We asked the participants not to disseminate the QR code to other people. Both surveys had four pages with about 20 items per page, which took about 20 min to complete. The Questionnaire Star tool performed a completeness check before participants submitted their questionnaires. Participants were able to review and change their responses. Upon completion, an electronic cash coupon of CNY10 (USD 1.5) was sent to participants as a token of appreciation. All data were stored on the online server of the survey platform and protected by a password. Only the corresponding author had access to the database.

We approached 2653 and 3060 eligible factory workers in the first and the second rounds, respectively; 600 and 434 of them refused to join the study due to a lack of time and other logistical reasons, respectively; and 2053 and 2626 completed the surveys, respectively (response rate: 77.3% in the first round and 85.8% in the second round). Among these participants, 844 (first round) and 1213 participants (second round) had at least one child aged 3–17 years. These parents answered additional questions about hesitancy and attitudes toward COVID-19 vaccination for their children. Ethics approval was obtained from the Longhua District CDC (references 2020001 and 2021015).

### 2.3. Measurement

#### 2.3.1. Questionnaire Development

A panel involving one CDC staff member, two public health researchers, a health psychologist, and a factory worker developed the questionnaires. The questionnaire was pilot tested among 10 factory workers to assess its clarity and readability. Participants in the pilot study believed that the length was acceptable and the contents were easy to comprehend. These 10 workers did not participate in the actual survey. The panel finalized the questionnaires. The measurements of the first round and second round of surveys were summarized in Table 1. We included the questionnaires in online Appendix A.

#### 2.3.2. Background Characteristics

In both surveys, participants reported sociodemographic information (i.e., age, gender, relationship status, education level, monthly personal income, whether they were frontline workers or management staff) and the age of their children. In the case of more than one child under the age of 18 years within their household, participants referred to the one whose birthday was closest to the survey date when answering questions. We used validated tools to measure compliance to five types of personal COVID-19 preventive measures in the past month in this study [50,51,52]. These preventive measures included their frequency of wearing facemasks when having close contact with others in the workplace and other public spaces and sanitizing their hands after returning from public spaces or touching public installations (response categories: every time, often, sometimes, and never). This study also measured whether they avoided social/meal gatherings with people who did not live together and in crowded places in the past month.

#### 2.3.3. COVID-19 Vaccine Hesitancy for Their Children

In the first round, COVID-19 vaccination was not yet available in China. We asked participants about their likelihood of letting their children be given a COVID-19 vaccination if available (response categories: 1—very unlikely, 2—unlikely, 3—neutral, 4—likely, and 5—very likely). In the second round, we first asked whether their children had received a COVID-19 vaccination. For parents whose children had not received COVID-19 vaccination, we asked their likelihood of letting their children be given a COVID-19 vaccination (response categories: 1—very unlikely, 2—unlikely, 3—neutral, 4—likely, and 5—very likely). Vaccine hesitancy was defined as “very unlikely”, “unlikely”, or “neutral” in both rounds. The same definition of vaccine hesitancy was used in published studies [46].

#### 2.3.4. Parents’ COVID-19 Vaccine Hesitancy

In the first round, we asked participants about their likelihood of having a COVID-19 vaccination if available (response categories: 1—very unlikely, 2—unlikely, 3—neutral, 4—likely, and 5—very likely). In the second round, we first asked whether they had received a COVID-19 vaccination. For participants who had not received a COVID-19 vaccination, we further asked their likelihood of receiving a COVID-19 vaccination. The definition of parents’ vaccine hesitancy was the same as vaccine hesitancy for their children.

#### 2.3.5. Attitudes toward COVID-19 Vaccination for Children

In both rounds, two scales measured positive attitudes and negative attitudes toward COVID-19 vaccination. The Cronbach’s alpha values of the 3-item Positive Attitude Scale and the 4-item Negative Attitude Scale were 0.71 and 0.64, respectively; single factors were identified via exploratory factor analysis, explaining 64.0% and 56.6% of the total variance. Two single items measured the perceived subjective norm (“Your family member will support you in having your child take up COVID-19 vaccination”) and perceived behavioral control (“Having your child receive COVID-19 vaccination is easy for you if you want them to”). Each item was rated as 1—disagree, 2—neutral, or 3—agree.

#### 2.3.6. Influence of Social Media

In both rounds of surveys, participants reported their frequency of exposure to experiences related to COVID-19 vaccination shared by recipients on social media (i.e., WeChat, WeChat Moments, Weibo, Tiktok) in the past month. In the second wave, participants were asked to report on the frequency of their exposure to four other types of information on the aforementioned social media. Such information included the following concerns: (1) the COVID-19 pandemic is not under control in countries after scaling up COVID-19 vaccination, (2) infectiousness and harms of the variants of concern of SARS-CoV-2, (3) outbreaks caused by variants of concern of SARS-CoV-2 in some places of China, and (4) people develop COVID-19 after receiving the primary series of COVID-19 vaccinations. The possible responses to these items were 1—almost none, 2—seldom, 3—sometimes, and 4—always.

### 2.4. Sample Size Planning

The target sample size for the original study was 2000 for each survey round. We explained the sample size planning for the original study in published papers [47,48]. We estimated that about 30% of the participants had a child aged 3–11 years (*n* = 600), and another 15% of them had a child aged 12–17 years (*n* = 300). Assuming that 20–30% of parents had vaccine hesitancy for their children in the first round, such a sample size could detect the smallest between-round difference in vaccine hesitancy of 6.1% among parents with a child aged 3–11 years and 8.3% among parents with a child aged 12–17 years (power = 0.80, alpha: 0.05; PASS 11.0, NCSS, LLC, Kaysville, UT, USA).

### 2.5. Statistical Analysis

The differences in background characteristics between parents of children aged 3–11 years in the first and the second rounds were compared using chi-square tests (for categorical variables) and independent-sample *t*-tests (for continuous variables). After controlling for background variables with significant differences between rounds, the difference in COVID-19 vaccine hesitancy and independent variables of interest (attitudes toward COVID-19 vaccination and influence of social media) were compared using logistic/linear regression models. The subsequent analysis was performed regarding parents in the same round of the survey. Using vaccine hesitancy for their children as the dependent variable, and background characteristics as independent variables, crude odds ratios (ORs) were obtained using logistic regression models. The associations between independent variables of interest and the dependent variable were then obtained by fitting a single logistic regression model involving one of the independent variables and all significant background characteristics. Adjusted odds ratios (AORs) and the respective 95% confidence intervals (CIs) were obtained. The same analysis was performed regarding parents of children aged 12–17 years. SPSS version 26.0 (IBM Corp., Armonk, NY, USA) was used for the data analysis, with *p* < 0.05 considered statistically significant.

## 3. Results

### 3.1. Background Characteristics

The mean age of parents of children aged 3–11 years was 34.8 years in the first round and 35.5 years in the second round. The majority of them were female (first round: 60%, second round: 57.9%), married (first round: 95.6%, second round: 95.8%), without tertiary education (first round: 75.1%, second round: 53.2%), with a monthly personal income between CNY 3000 and CNY 6999 (USD 462–1077) (first round: 61.9%, second round: 60.8%), and working as frontline workers (first round: 66.1%, second round: 61.5%). As compared with parents in the first round, those in the second round were slightly older (*p* = 0.01) and more likely to have tertiary education (*p* < 0.001) and a higher income (*p* < 0.001). Regarding parents of children aged 12–17 years, those in the second round were slightly younger (40.7 years versus 41.7 years, *p* = 0.01), more likely to be male (43.5% versus 27.2%, *p* < 0.001), and have received higher education (tertiary: 40.0% versus 15.4%, *p* < 0.001) and a higher income level (≥CNY 7000: 26.8% versus 7.9%, *p* < 0.001) compared with those in the first round (Table 2).

### 3.2. Changes in Parents’ COVID-19 Vaccine Hesitancy

The prevalence of vaccine hesitancy for children aged 3–11 years was 25.9% in the first round and 17.4% in the second round, while such a prevalence for children aged 12–17 years dropped from 26.0% in the first round to 3.5% in the second round. The decline in vaccine hesitancy for children was statistically significant in both sub-groups of parents (*p* < 0.001). In the second round, 12.8% of children aged 3–11 years and 85.6% of those aged 12–17 years received at least one dose of COVID-19 vaccination (Table 3 and Table 4).

In the first round, there was no difference in vaccine hesitancy for children between these two sub-groups of parents (*p* = 0.90). However, parents of children aged 12–17 years reported significantly lower vaccine hesitancy compared with parents of children aged 3–11 years in the second round (*p* < 0.001).

### 3.3. Changes in Attitudes toward COVID-19 Vaccination for Children and Information Exposure on Social Media

In both groups of parents, significant increases in positive attitudes, negative attitudes, the perceived subjective norm, and the perceived behavioral control related to children’s COVID-19 vaccination were observed when comparing the second round with the first round. Both sub-groups of parents were exposed to testimonials given by COVID-19 vaccination recipients on social media more frequently in the second round than in the first round (Table 3 and Table 4).

### 3.4. Factors Associated with COVID-19 Vaccine Hesitancy among Parents of Children Aged 3–11 Years

Among this group of parents of children, better compliance with physical distancing behaviors was associated with lower vaccine hesitancy for children in the first round. A higher education level and wearing a face mask in the workplace were associated with lower vaccine hesitancy for children in the second round (Table 5). After adjusting for these significant background characteristics, positive attitudes (AOR: 0.53 and 0.48, *p* < 0.001), perceived higher support from significant others (AOR: 0.21 and 0.25, *p* < 0.001), and better behavioral control related to children’s COVID-19 vaccination (AOR: 0.57 and 0.39, *p* < 0.001) were associated with lower vaccine hesitancy for children in the first and second rounds, respectively. COVID-19 vaccine hesitancy among parents was associated with higher vaccine hesitancy for children (AOR: 2.50 and 6.34, *p* < 0.001 and *p* = 0.01) in the first and second rounds, respectively. Negative attitudes toward COVID-19 vaccination for children (AOR: 1.11, *p* = 0.03) and frequency of exposure to experiences related to COVID-19 vaccination shared by recipients on social media (AOR: 0.80, *p* = 0.02) were associated with vaccine hesitancy in the second round, but not in the first round. In the second round, a higher frequency of exposure to information on infectiousness and harms of variants of concern was associated with lower vaccine hesitancy (AOR: 0.83, *p* = 0.04), while exposure to information on developing COVID-19 after receiving a primary COVID-19 vaccination series was associated with higher vaccine hesitancy (AOR: 1.24, *p* = 0.04) (Table 6).

### 3.5. Factors Associated with COVID-19 Vaccine Hesitancy among Parents of Children Aged 12–17 Years

Among the parents of children aged 12–17 years, better compliance with face mask wearing in public spaces other than the workplace was associated with lower vaccine hesitancy for children in both rounds. In the second round, older age was associated with lower vaccine hesitancy, while being single or divorced was associated with higher vaccine hesitancy (Table 7). After adjusting for these significant background characteristics, factors associated with vaccine hesitancy for children were the same between the two rounds. These factors were positive attitudes (AOR: 0.71 and 0.55, *p* = 0.002 and 0.01), perceived higher support from significant others (AOR: 0.33 and 0.26, *p* < 0.001 and *p* = 0.002), and better behavioral control related to children’s COVID-19 vaccination (AOR: 0.51 and 0.42, *p* = 0.003 and 0.049) for the first and second round, respectively (Table 8).

## 4. Discussion

This was one of the first studies that tracked changes in parents’ COVID-19 vaccine hesitancy for children. This repeated cross-sectional survey had the strengths of considering associated factors at both individual and interpersonal levels, as well as a relatively large sample size. The results showed that Chinese parents were responsive to the national childhood COVID-19 vaccination programs. The prevalence of vaccine hesitancy for children decreased significantly in both sub-groups of parents after the rollout of the vaccination programs. As compared with parents with children aged 3–11 years, a larger decrease in vaccine hesitancy was observed among parents with children aged 12–17 years (22.5% versus 8.5%). The rollout of the national vaccination program for children aged 12–17 years was 3 months earlier than that for younger children, which might partially explain the difference. The level of parental vaccine hesitancy after the rollout of the national childhood COVID-19 vaccination program was similar to that reported in Brazil (9%) [53] but was lower than that reported in Germany (49%) [54] and Italy (48.3%) [55]. Regarding the uptake rate, 85% of children aged 12–17 years received at least one dose of a COVID-19 vaccination. Such coverage was lower than that reported in Singapore (98%) [22] but was higher than those observed in the United States (59%) and the United Kingdom (24%) [23,24]. Although the program for children aged 3–11 years had just started, 12.8% of children in this age group had already received a COVID-19 vaccination. Given the low vaccine hesitancy among parents, the coverage of COVID-19 vaccination in children aged 3–11 years is expected to grow substantially.

As compared with the first round, attitudes favoring children’s COVID-19 vaccination (i.e., positive attitudes, a perceived subjective norm, and perceived behavioral control) increased significantly in both groups of parents in the second round. The availability of vaccines for children and health promotion efforts done by the government might have caused such positive changes since these attitudes were associated with lower vaccine hesitancy for children in both rounds. Similar to studies conducted in other countries [53,54,55], positive attitudes toward children’s COVID-19 vaccination were associated with lower vaccine hesitancy in both rounds. Changes in such attitudes might have contributed to the decrease in vaccine hesitancy over time. Such findings provided empirical implications to inform health promotion. Health communication messages should emphasize vaccine efficacy for children, the importance of childhood vaccination in COVID-19 control, and the adequate supply of vaccines in China. These messages may strengthen their positive attitudes toward children’s COVID-19 vaccination. Future programs should also encourage parents to discuss children’s vaccination with other family members to obtain more support from their significant others. Simplifying the procedures for children to receive COVID-19 vaccination may be helpful to increase perceived behavioral control among parents.

In the first round, relatively few parents had negative attitudes toward their child receiving a COVID-19 vaccination. Negative attitudes were not barriers to children’s COVID-19 vaccination uptake at this time point. However, concerns about the side effects, the duration of protection, and time constraints regarding taking the children to receive COVID-19 vaccination increased significantly in both groups of parents in the second round. Negative attitudes toward children’s COVID-19 vaccination were associated with higher vaccine hesitancy among parents with children aged 3–11 years. Concerns about safety and efficacy were major contributors to parents’ hesitancy to vaccinate their children against diseases other than COVID-19 (e.g., human papillomavirus, measles, mumps, and rubella) [56]. Since the majority of the adults in China received COVID-19 vaccinations, parents learned more about the potential side effects from their or others’ experiences. Understandably, they worried about potential unknown long-term effects and serious side effects of the vaccines for their children [57]. Up-to-date information about the safety of COVID-19 vaccines should be disseminated to parents to reduce such concerns. Emerging evidence showed waning protection against COVID-19 after receiving a two-dose primary vaccine series [58,59,60,61,62]. Since the COVID-19 booster dose was not recommended for children in the second round, the short duration of protection became the largest concern among parents. For adolescents, a booster dose was associated with increased COVID-19 vaccine efficacy [63]. International authorities recommend a COVID-19 booster dose for children [64,65]. In Hong Kong, China, the government started recommending and offering a booster dose for children aged 3 years or above on January 2022 [66]. Therefore, China may consider updating the vaccination guidelines and offer a booster dose for children to reduce parents’ concerns about the duration of protection. Implementing school-based COVID-19 vaccination programs may be helpful to reduce parents’ concerns about the lack of time to take their children to receive vaccines. Our findings were similar to those observed by a previous study that showed that preventive health behavior is predominately influenced by the consumers’ efficacy (e.g., perceived benefits and barriers of the preventive behavior) and social effects (e.g., subjective norm), but not the perceived threat [67]. Such findings would contribute to the marketing of childhood COVID-19 vaccination programs.

In the first round, few parents were exposed to experiences related to COVID-19 vaccination shared by recipients on social media. Such exposure was not associated with vaccine hesitancy for their children. In the second round, both groups of parents reported high exposure to such testimonials on social media. Such exposure was associated with lower vaccine hesitancy among parents with children aged 3–11 years at this time point. The emergence of SARS-CoV-2 variants of concern also had an impact on parents’ vaccine hesitancy. Higher exposure to information about infectiousness and harms of the SARS-CoV-2 variants of concern was also associated with lower vaccine hesitancy. The perceived higher threat of the variants of concern was associated with a higher intention to receive COVID-19 vaccination in the Chinese population [47]. Health authorities in China should consider using their official social media accounts to disseminate health communication messages, as Chinese parents considered such channels to be credible information sources [51]. Higher exposure to information on developing COVID-19 after receiving a primary vaccine series was associated with higher vaccine hesitancy for their children. Parents with such exposure might doubt the effectiveness of the COVID-19 vaccines. Health communication messages should explain that although the vaccine efficacy at preventing COVID-19 from developing declined over time, completing the primary vaccine series is still effective at preventing severe consequences and deaths [68]. Such findings highlight the need for a COVID-19 vaccine booster dose for children in China.

This study had some limitations. First, it was not possible to ascertain whether some parents answered both the first and second rounds of the survey, as we did not ask participants in the second round whether they attended the first round. We recruited factory workers from organizations that provided physical examinations. Factory workers in Shenzhen are required to receive a physical examination once every 12 months to renew their working permit/contract. The interval between the two rounds of surveys was 14 months in this study. It was likely that most of the participants in the first round had completed their physical examination before the second round. Therefore, the chance of having the same participant completing both rounds was very low. Second, we did not measure the awareness of a childhood COVID-19 vaccination campaign among participants in the second round. China has been actively promoting childhood COVID-19 vaccination through all mass media channels. The schools in China also actively disseminated information about the campaign to the parents. Therefore, we believed the awareness of such a campaign should be very high among parents. Third, we did not ask for a history of SARS-CoV-2 infection, which influenced vaccine hesitancy in a previous study [69]. Previous studies suggested that COVID-19 survivors are stigmatized in China [70,71]. Therefore, we decided to omit any questions on the participant’s history of COVID-19 in the questionnaires. Since China has been implementing the zero COVID policy during the study period and the daily confirmed COVID-19 cases were low, the prevalence of SARS-CoV-2 infection was very low in factory workers. Therefore, the impact of a history of COVID-19 on vaccine hesitancy might be limited among Chinese parents. Fourth, we did not include some potential facilitators/barriers due to the limited length of our questionnaires, such as the fear of restrictions for non-vaccinated children. Fifth, we only included parents who were factory workers, as this was a secondary analysis. Over 30% of Shenzhen’s population were factory workers [49]. However, the failure to include parents of other occupations or those without full-time jobs limited the representativeness of our study samples. As compared with parents with other occupations (e.g., healthcare workers) [28] and those from the general population [27], the level of vaccine hesitancy for children was lower among parents who were factory workers. Therefore, our results might underestimate the level of COVID-19 vaccine hesitancy for children among parents in Shenzhen. Moreover, we only recruited participants in one Chinese city. Since Shenzhen is one of the most developed cities in China, the findings might not be generalizable to less developed regions in China. In China, COVID-19 vaccination was first implemented in large cities, such as Shenzhen, and the local government has spent a great amount of effort and resources to promote COVID-19 vaccination. It was likely that the level of vaccine hesitancy was lower among parents in Shenzhen as compared with other less developed regions. Sixth, since the study was anonymous and did not collect participants’ identification information, we were not able to collect information from refusals. People who refused to join the study might have different characteristics compared with the study participants; therefore, it is reasonable to believe that selection bias existed. However, the response rate for both rounds of surveys was higher than other online surveys for similar topics [27,28,29,30,31,32,33,34,35,36,37,38,39,40]. The response rate was significantly higher in the second round as compared with the first round. The data collection method was identical between these two rounds of surveys. However, participants in the second round had a higher education level. It is possible that those with higher education would find it easier to complete an online survey, and thus, leading to a higher response rate. In our study, we controlled the education level when comparing our outcome variables. Moreover, data were self-reported, and verification was not feasible. Participants might under-report vaccine hesitancy due to social desirability. Furthermore, respondents without vaccine hesitancy might not necessarily translate into actual uptake. They might still delay their children’s COVID-19 vaccination. Last but not the least, we could not provide evidence about the causal associations between changes in contextual factors (e.g., pandemic, policies) and changes in vaccine hesitancy. The cross-sectional design could not establish causality. Future studies should monitor the changes in COVID-19 vaccination hesitancy and actual uptake for children among parents. These studies should include parents of other occupations or those without full-time jobs, as well as involve study sites in less developed regions in China. Further investigation of behavioral control should be helpful to develop strategies to overcome parental vaccine hesitancy [72].

## 5. Conclusions

The prevalence of parents’ vaccine hesitancy decreased significantly after the rollout of the national childhood COVID-19 vaccination programs in China, especially among parents with children aged 12–17 years. Positive attitudes, negative attitudes, perceived support from significant others, perceived behavioral control related to children’s COVID-19 vaccination, and social media exposure related to COVID-19 vaccination increased significantly over time. Different factors influenced parents’ vaccine hesitancy before and after the vaccination program rollout. Regular monitoring of vaccine hesitancy and its associated factors among parents should be conducted to guide health promotion and policy making.

## Figures and Tables

**Figure 1 vaccines-10-01478-f001:**
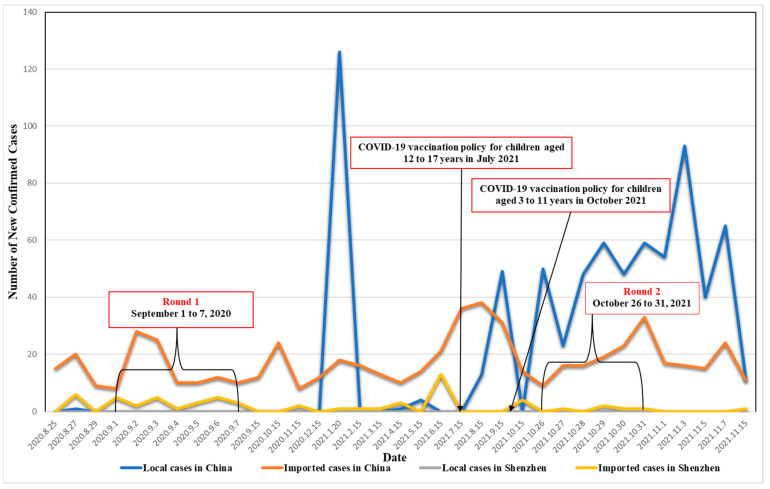
The COVID-19 situation and policy changes related to COVID-19 vaccination during the study period in China and Shenzhen.

**Table 1 vaccines-10-01478-t001:** Measurements in the first and second rounds of the surveys.

Measurements	Round 1	Round 2
**Sociodemographic characteristics**		
Age	√	√
Gender	√	√
Relationship status	√	√
Education level	√	√
Monthly personal income, CNY (USD)	√	√
Type of work	√	√
**Personal COVID-19 preventive measures in the past month**		
Frequency of face mask wearing in public spaces or on transportation other than in the workplace	√	√
Frequency of face mask wearing when you have close contact with other people in the workplace	√	√
Frequency of sanitizing of hands by using soaps, liquid soaps, or alcohol-based sanitizer after returning from public spaces or touching public installations	√	√
Self-reported avoidance of social and meal gatherings with other people who do not live together	√	√
Self-reported avoiding of crowded places	√	√
**COVID-19 vaccination uptake and hesitancy**		
Number of doses of COVID-19 vaccination received by their children	ⅹ	√
Likelihood of letting their children be given a COVID-19 vaccination ^1^	√	√
**Perceptions related to COVID-19 vaccination**		
Positive attitudes		
COVID-19 vaccination is highly effective in protecting your child from COVID-19	√	√
COVID-19 vaccination can contribute to the control of COVID-19 in China	√	√
China will have an adequate supply of COVID-19 vaccines	√	√
Negative attitudes	√	√
Your child will have severe side effects after receiving a COVID-19 vaccination	√	√
The protection of COVID-19 vaccines will only last for a short time	√	√
Your child is afraid of vaccination	√	√
You do not have time to take your child for a COVID-19 vaccination	√	√
Perceived subjective norm related to child’s COVID-19 vaccination: your family member would support you in letting the child be given a COVID-19 vaccination	√	√
Perceived behavioral control to let the child be given a COVID-19 vaccination: having the child receive COVID-19 vaccination is easy for you if you want them to	√	√
**Frequency of exposure to the following information on social media (e.g., WeChat, WeChat moments, Weibo, Tiktok) in the past month**		
Experiences related to COVID-19 vaccination shared by recipients on social media	√	√
The COVID-19 pandemic is not under control in some countries after scaling up COVID-19 vaccination	ⅹ	√
Infectiousness and harms of the variants of concern of SARS-CoV-2	ⅹ	√
Outbreaks caused by variants of concern of SARS-CoV-2 in some places in China	ⅹ	√
People develop COVID-19 after receiving a primary series of COVID-19	ⅹ	√

^1^ Only the parents whose children had not yet received a COVID-19 vaccination answered this question.

**Table 2 vaccines-10-01478-t002:** Background characteristics of the parents.

Characteristics	Parents of Children Aged 3–11 Years			Parents of Children Aged 12–17 Years		
	Round 1 (*n* = 590)	Round 2 (*n* = 873)		Round 1 (*n* = 254)	Round 2 (*n* = 340)	
	*n* (%)	*n* (%)	*p*-Values ^1^	*n* (%)	*n* (%)	*p*-Values ^1^
**Sociodemographic characteristics**						
Age, years, mean (SD)						
	34.8 (5.3)	35.5 (5.1)	0.01	41.7 (4.7)	40.7 (5.2)	0.01
Gender						
Male	236 (40.0)	387 (44.3)		69 (27.2)	148 (43.5)	
Female	354 (60.0)	486 (57.9)	0.10	185 (72.8)	192 (56.5)	<0.001
Relationship status						
Married	564 (95.6)	836 (95.8)		238 (93.7)	328 (96.5)	
Single or divorced	24 (4.1)	32 (3.7)		16 (6.3)	11 (3.2)	
Having a stable partner	2 (0.3)	5 (0.6)	0.76	0 (0.0)	1 (0.3)	0.15
Education level						
Senior high or below	443 (75.1)	464 (53.2)		215 (84.6)	204 (60.0)	
College and above	147 (24.9)	409 (46.8)	<0.001	39 (15.4)	136 (40.0)	<0.001
Monthly personal income, CNY (USD)						
<3000 (462)	128 (21.7)	93 (10.7)		61 (24.0)	47 (13.8)	
3000–6999 (462–1077)	365 (61.9)	531 (60.8)		173 (68.1)	202 (59.4)	
≥7000 (1078)	97 (16.4)	249 (28.5)	<0.001	20 (7.9)	91 (26.8)	<0.001
Type of work						
Frontline workers	390 (66.1)	537 (61.5)		179 (70.5)	226 (66.5)	
Management staff	200 (33.9)	336 (38.5)	0.07	75 (29.5)	114 (33.5)	0.30
**Personal COVID-19 preventive measures in the past month**						
Frequency of face mask wearing in public spaces or on transportation other than in the workplace						
Every time	495 (83.9)	746 (85.5)		210 (82.7)	261 (76.8)	
Often	78 (13.2)	101 (11.6)		32 (12.6)	64 (18.8)	
Sometimes	17 (2.9)	26 (3.0)		12 (4.7)	14 (4.1)	
Never	0 (0.0)	0 (0.0)	0.64	0 (0.0)	1 (0.3)	0.17
Frequency of face mask wearing when you have close contact with other people in the workplace						
Every time	416 (70.5)	643 (73.7)		224 (88.2)	302 (88.8)	
Often	127 (21.5)	160 (18.3)		27 (10.6)	32 (9.4)	
Sometimes	45 (7.6)	62 (7.1)		3 (1.2)	6 (1.8)	
Never	2 (0.3)	8 (0.9)	0.24	0 (0.0)	0 (0.0)	0.76
Frequency of sanitizing of hands by using soaps, liquid soaps, or alcohol-based sanitizer after returning from public spaces or touching public installations						
Every time	352 (41.4)	484 (55.4)		154 (60.6)	190 (55.9)	
Often	155 (26.3)	206 (23.6)		69 (27.2)	88 (25.9)	
Sometimes	87 (14.7)	165 (18.9)		28 (11.0)	53 (15.6)	
Never	6 (1.0)	18 (2.1)	0.06	3 (1.2)	6 (2.6)	0.22
Self-reported avoidance of social and meal gatherings with other people who do not live together						
No	241 (40.8)	362 (41.5)		107 (42.1)	139 (40.9)	
Yes	349 (59.2)	511 (58.5)	0.81	147 (57.9)	201 (59.1)	0.76
Self-reported avoidance of crowded places						
No	205 (34.7)	293 (33.6)		78 (30.7)	119 (34.7)	
Yes	385 (65.3)	580 (66.4)	0.64	176 (69.3)	222 (65.3)	0.31

^1^ Comparing round 1 and round 2 using chi-square tests.

**Table 3 vaccines-10-01478-t003:** Changes in uptake, hesitancy, and attitudes related to COVID-19 vaccination for children aged 3–11 years.

	Round 1 (*n* = 590)	Round 2 (*n* = 873)	OR (95% CI) *p*-Value	AOR (95% CI) *p*-Value
**COVID-19 vaccination uptake and hesitancy**				
Number of doses of COVID-19 vaccination received by their children, *n* (%)				
0	0 (0.0)	761 (87.2)		
1	0 (0.0)	37 (4.2)		
2	0 (0.0)	75 (8.6)	N.A.	N.A.
COVID-19 vaccine hesitancy for their children, *n* (%)				
No	437 (74.1)	721 (82.6)	0.60 (0.47, 0.78)	0.57 (0.44, 0.75)
Yes	153 (25.9)	152 (17.4)	*p* < 0.001	*p* < 0.001
COVID-19 vaccine hesitancy among parents, *n* (%)				
No	466 (79.0)	864 (99.0)	0.04 (0.02, 0.08)	0.04 (0.02, 0.09)
Yes	124 (21.0)	9 (1.0)	*p* < 0.001	*p* < 0.001
**Attitudes toward COVID-19 vaccination for their children**				
Positive attitudes toward COVID-19 vaccination, *n* (%) agree				
COVID-19 vaccination is highly effective in protecting your child from COVID-19	334 (56.6)	614 (70.3)	1.82 (1.46, 2.26) *p* < 0.001	1.76 (1.41, 2.21) *p* < 0.001
COVID-19 vaccination can contribute to the control of COVID-19 in China	507 (85.9)	808 (92.6)	2.04 (1.44, 2.87) *p* < 0.001	1.75 (1.23, 2.50) *p* = 0.002
China will have an adequate supply of COVID-19 vaccines	432 (73.2)	788 (90.3)	3.39 (2.54, 4.53) *p* < 0.001	2.94 (2.18, 3.96) *p* < 0.001
Positive Attitude Scale score, mean (SD)	8.1 (1.2)	8.5 (0.9)	B (95% CI) 0.39 (0.28, 0.50), *p* < 0.001	Adjusted B (95% CI) 0.34 (0.23, 0.45), *p* < 0.001
Negative attitudes toward COVID-19 vaccination, *n* (%) agree				
Your child will have severe side effects after receiving a COVID-19 vaccination	44 (7.5)	131 (15.0)	2.19 (1.53, 3.14) *p* < 0.001	2.26 (1.56, 3.27) *p* < 0.001
The protection of COVID-19 vaccines will only last for a short time	116 (19.7)	430 (49.3)	3.97 (3.11, 5.06) *p* < 0.001	4.06 (3.16, 5.22) *p* < 0.001
Your child is afraid of vaccination	118 (20.0)	206 (23.6)	1.24 (0.96, 1.59) *p* = 0.10	1.24 (0.95, 1.62) *p* = 0.11
You do not have time to take your child for a COVID-19 vaccination	138 (23.4)	250 (28.6)	1.31 (1.03, 1.67) *p* = 0.03	1.43 (1.11, 1.83) *p* = 0.005
Negative Attitude Scale score, mean (SD)	7.7 (1.6)	8.0 (1.9)	B (95% CI) 0.29 (0.10, 0.47), *p* = 0.003	Adjusted B (95% CI) 0.35 (0.15, 0.54), *p* < 0.001
Perceived subjective norm related to child’s COVID-19 vaccination: your family member would support you in letting the child be given a COVID-19 vaccination				
Agree, *n* (%)	300 (50.8)	615 (70.4)	2.30 (1.85, 2.86) *p* < 0.001	2.25 (1.79, 2.82) *p* < 0.001
Response score, mean (SD)	2.5 (0.6)	2.6 (0.6)	B (95% CI) 0.18 (0.12, 0.24), *p* < 0.001	Adjusted B (95% CI) 0.17 (0.11, 0.23), *p* < 0.001
Perceived behavioral control to let the child be given a COVID-19 vaccination: having the child receive COVID-19 vaccination is easy for you if you want them to				
Agree, *n* (%)	260 (44.1)	589 (67.5)	2.63 (2.12, 3.27) *p* < 0.001	2.43 (1.94, 3.04) *p* < 0.001
Response score, mean (SD)	2.3 (0.7)	2.6 (0.6)	B (95% CI) 0.27 (0.21, 0.34), *p* < 0.001	Adjusted B (95% CI) 0.24 (0.17, 0.31), *p* < 0.001
**Frequency of exposure to the following information on social media (e.g., WeChat, WeChat moments, Weibo, Tiktok) in the past month**				
Experiences related to COVID-19 vaccination shared by recipients on social media, *n* (%)				
Almost none	278 (47.1)	199 (22.8)		
Seldom	156 (26.4)	299 (34.2)		
Sometimes	106 (18.0)	261 (29.9)		
Always	50 (8.5)	114 (13.1)		
Response score, mean (SD)	1.9 (1.0)	2.3 (1.0)	B (95% CI) 0.45 (0.35, 0.56), *p* < 0.001	Adjusted B (95% CI) 0.47 (0.37, 0.58), *p* < 0.001
The COVID-19 pandemic is not under control in some countries after scaling up COVID-19 vaccination				
Almost none		182 (20.8)		
Seldom		271 (31.0)		
Sometimes		261 (29.9)		
Always	N.A.	159 (18.2)		
Response score, mean (SD)	N.A.	2.5 (1.0)	N.A.	N.A.
Infectiousness and harms of the variants of concern of SARS-CoV-2				
Almost none		114 (13.1)		
Seldom		217 (24.9)		
Sometimes		280 (32.1)		
Always	N.A.	262 (30.0)		
Response score, mean (SD)	N.A.	2.8 (1.0)	N.A.	N.A.
Outbreaks caused by variants of concern of SARS-CoV-2 in some places in China				
Almost none		130 (14.9)		
Seldom		293 (33.6)		
Sometimes		298 (34.1)		
Always	N.A.	152 (17.4)		
Response score, mean (SD)	N.A.	2.5 (0.9)	N.A.	N.A.
People develop COVID-19 after receiving the primary series of COVID-19 vaccinations				
Almost none		181 (20.7)		
Seldom		385 (44.1)		
Sometimes		237 (27.1)		
Always	N.A.	70 (8.0)		
Response score, mean (SD)	N.A.	2.2 (0.9)	N.A.	N.A.

OR: crude odds ratio. AOR: adjusted odds ratio, where the odds ratio was adjusted for age, education level, and income level. B: unstandardized coefficient. Adjusted B: unstandardized coefficient adjusted for age, education level, and income level. CI: confidence interval. N.A.: not applicable.

**Table 4 vaccines-10-01478-t004:** Changes in uptake, hesitancy, and attitudes related to COVID-19 vaccination for children aged 12–17 years.

	Round 1 (*n* = 254)	Round 2 (*n* = 340)	OR (95% CI) *p*-Value	AOR (95% CI) *p*-Value
**COVID-19 vaccination uptake and hesitancy**				
Number of doses of COVID-19 vaccination received by their children, *n* (%)				
0	0 (0.0)	49 (14.4)		
1	0 (0.0)	31 (9.1)		
2	0 (0.0)	260 (76.5)	N.A.	N.A.
COVID-19 vaccine hesitancy for their children, *n* (%)				
No	188 (74.0)	328 (96.5)	0.10 (0.06, 0.20)	0.10 (0.05, 0.20)
Yes	66 (26.0)	12 (3.5)	*p* < 0.001	*p* < 0.001
COVID-19 vaccine hesitancy among parents, *n* (%)				
No	195 (76.8)	337 (99.1)	0.03 (0.01, 0.10)	0.03 (0.01, 0.11)
Yes	59 (23.2)	3 (0.9)	*p* < 0.001	*p* < 0.001
**Attitudes toward COVID-19 vaccination for their children**				
Positive attitudes toward COVID-19 vaccination, *n* (%) agree				
COVID-19 vaccination is highly effective in protecting your child from COVID-19	150 (59.1)	255 (75.0)	2.08 (1.47, 2.95) *p* < 0.001	2.11 (1.45, 3.07) *p* < 0.001
COVID-19 vaccination can contribute to the control of COVID-19 in China	209 (82.3)	315 (92.6)	2.71 (1.61, 4.56) *p* < 0.001	2.78 (1.59, 4.88) *p* < 0.001
China will have an adequate supply of COVID-19 vaccines	170 (66.9)	303 (89.1)	4.05 (2.63, 6.22) *p* < 0.001	3.63 (2.29, 5.74) *p* < 0.001
Positive Attitude Scale score, mean (SD)	7.9 (1.3)	8.5 (0.9)	B (95% CI) 0.57 (0.39, 0.74) *p* < 0.001	Adjusted B (95% CI) 0.52 (0.34, 0.71) *p* < 0.001
Negative attitudes toward COVID-19 vaccination, *n* (%) agree				
Your child will have severe side effects after receiving a COVID-19 vaccination	33 (13.0)	44 (12.9)	0.99 (0.61, 1.62) *p* = 0.99	0.97 (0.58, 1.62) *p* = 0.90
The protection of COVID-19 vaccines will only last for a short time	49 (19.3)	161 (47.4)	3.76 (2.58, 5.49) *p* < 0.001	4.03 (2.70, 6.02) *p* < 0.001
Your child is afraid of vaccination	49 (19.3)	61 (17.9)	0.92 (0.60, 1.39) *p* = 0.68	0.93 (0.60, 1.45) *p* = 0.74
You do not have time to take your child for a COVID-19 vaccination	63 (24.8)	113 (33.2)	1.51 (1.05, 2.17) *p* = 0.03	1.50 (1.02, 2.21) *p* = 0.04
Negative Attitude Scale score, mean (SD)	7.8 (1.6)	7.4 (2.0)	B (95% CI) −0.40 (−0.71, −0.09), *p* = 0.01	Adjusted B (95% CI) −0.39 (−0.72, −0.06), *p* = 0.02
Perceived subjective norm related to child’s COVID-19 vaccination: your family member would support you in letting the child be given a COVID-19 vaccination				
Agree, *n* (%)	137 (53.9)	302 (88.8)	6.79 (4.47, 10.31) *p* < 0.001	7.09 (4.51, 11.15) *p* < 0.001
Response score, mean (SD)	2.5 (0.6)	2.9 (0.5)	B (95% CI)	Adjusted B (95% CI)
			0.36 (0.27, 0.44) *p* < 0.001	0.37 (0.28, 0.46) *p* < 0.001
Perceived behavioral control to let the child be given a COVID-19 vaccination: having the child receive COVID-19 vaccination is easy for you if you want them to				
Agree, *n* (%)	97 (38.2)	273 (80.3)	6.60 (4.56, 9.53) *p* < 0.001	6.10 (4.12, 9.06) *p* < 0.001
Response score, mean (SD)	2.3 (0.7)	2.8 (0.5)	B (95% CI)	Adjusted B (95% CI)
			0.49 (0.40, 0.59) *p* < 0.001	0.45 (0.35, 0.55) *p* < 0.001
**Frequency of exposure to the following information on social media (e.g., WeChat, WeChat moments, Weibo, Tiktok) in the past month**				
Experiences related to COVID-19 vaccination shared by recipients on social media, *n* (%)				
Almost none	125 (49.2)	82 (24.1)		
Seldom	55 (21.7)	116 (34.1)		
Sometimes	46 (18.1)	101 (29.7)		
Always	28 (11.0)	41 (12.1)		
Response score, mean (SD)	1.9 (1.1)	2.3 (1.0)	B (95% CI)	Adjusted B (95% CI)
			0.39 (0.22, 0.55) *p* < 0.001	0.44 (0.27, 0.62) *p* < 0.001
The COVID-19 pandemic is not under control in some countries after scaling up COVID-19 vaccination				
Almost none		53 (15.6)		
Seldom		108 (31.8)		
Sometimes		106 (31.2)		
Always	N.A.	73 (21.5)		
Response score, mean (SD)	N.A.	2.6 (1.0)	N.A.	N.A.
Infectiousness and harms of the variants of concern of SARS-CoV-2				
Almost none		37 (10.9)		
Seldom		92 (27.1)		
Sometimes		116 (34.1)		
Always	N.A.	95 (27.9)		
Response score, mean (SD)	N.A.	2.8 (1.0)	N.A.	N.A.
Outbreaks caused by variants of concern of SARS-CoV-2 in some places in China				
Almost none		45 (13.2)		
Seldom		128 (37.6)		
Sometimes		119 (35.0)		
Always	N.A.	48 (14.1)		
Response score, mean (SD)	N.A.	2.5 (0.9)	N.A.	N.A.
People develop COVID-19 after receiving primary series of COVID-19				
Almost none		70 (20.6)		
Seldom		165 (48.5)		
Sometimes		86 (25.3)		
Always	N.A.	19 (5.6)		
Response score, mean (SD)	N.A.	2.2 (0.8)	N.A.	N.A.

OR: crude odds ratio. AOR: adjusted odds ratio, where the odds ratio was adjusted for age, education level, and income level. B: unstandardized coefficient. Adjusted B: unstandardized coefficient adjusted for age, education level, and income level. CI: confidence interval. N.A.: not applicable.

**Table 5 vaccines-10-01478-t005:** Associations between background characteristics and COVID-19 vaccine hesitancy for children aged 3–11 years.

	Round 1 (*n* = 590)		Round 2 (*n* = 873)	
	OR (95% CI)	*p*-Value	OR (95% CI)	*p*-Value
**Sociodemographic** **characteristics**				
Age, years	1.00 (0.96, 1.03)	0.78	0.96 (0.93, 1.00)	0.051
Gender				
Male	1.0		1.0	
Female	1.13 (0.77, 1.64)	0.54	0.98 (0.69, 1.39)	0.91
Relationship status				
Married	1.0		1.0	
Single or divorced	0.74 (0.27, 2.02)	0.56	0.88 (0.33, 2.33)	0.80
Having a stable partner	N.A.	N.A.	3.18 (0.53, 19.18)	0.21
Education level				
Senior high or below	1.0		1.0	
College and above	1.09 (0.72, 1.66)	0.68	1.46 (1.03, 2.07)	0.04
Monthly personal income, CNY (USD)				
<3000 (462)	1.0		1.0	
3000–6999 (462–1077)	1.10 (0.70, 1.74)	0.68	0.89 (0.50, 1.60)	0.70
≥7000 (1078)	0.70 (0.37, 1.33)	0.28	1.30 (0.70, 2.42)	0.40
Type of work				
Frontline workers	1.0		1.0	
Management staff	1.05 (0.71, 1.54)	0.82	1.20 (0.84, 1.71)	0.31
**Personal COVID-19 preventive measures in the past month**				
Frequency of face mask wearing in public spaces or on transportation other than in the workplace				
Often/sometimes/never	1.0		1.0	
Every time	0.72 (0.44, 1.16)	0.17	0.75 (0.47, 1.19)	0.22
Frequency of face mask wearing when you have close contact with other people in the workplace				
Often/sometimes/never	1.0		1.0	
Every time	0.78 (0.53, 1.16)	0.23	0.59 (0.40, 0.85)	0.01
Frequency of sanitizing of hands by using soaps, liquid soaps, or alcohol-based sanitizer after returning from public spaces or touching public installations				
Often/sometimes/never	1.0		1.0	
Every time	0.91 (0.63, 1.32)	0.61	0.72 (0.51, 1.02)	0.07
Self-reported avoidance of social and meal gatherings with other people who do not live together				
No	1.0		1.0	
Yes	0.57 (0.39, 0.83)	0.003	0.77 (0.54, 1.10)	0.15
Self-reported avoidance of crowded places				
No	1.0		1.0	
Yes	0.55 (0.38, 0.80)	0.002	0.76 (0.53, 1.09)	0.13

OR: crude odds ratio. CI: confidence interval. N.A.: not applicable.

**Table 6 vaccines-10-01478-t006:** Factors associated with vaccine hesitancy for children aged 3–11 years.

	Round 1 (*n* = 590)		Round 2 (*n* = 873)	
	AOR (95% CI)	*p*-Value	AOR (95% CI)	*p*-Value
COVID-19 vaccine hesitancy among parents				
No	1.0		1.0	
Yes	2.50 (1.63, 3.81)	<0.001	6.34 (1.66, 24.22)	0.01
**Attitudes toward children’s COVID-19 vaccination**				
Positive Attitude Scale	0.53 (0.45, 0.63)	<0.001	0.48 (0.40, 0.57)	<0.001
Negative Attitude Scale	1.05 (0.94, 1.19)	0.40	1.11 (1.01, 1.21)	0.03
Perceived subjective norm related to child’s COVID-19 vaccination	0.21 (0.14, 0.30)	<0.001	0.25 (0.19, 0.33)	<0.001
Perceived behavioral control to let the child be given a COVID-19 vaccination	0.57 (0.43, 0.76)	<0.001	0.39 (0.30, 0.50)	<0.001
**Frequency of exposure to the following information on social media (e.g., WeChat, WeChat moments, Weibo,** **Tiktok) in the past month**				
Experiences related to COVID-19 vaccination shared by recipients on social media	0.97 (0.80, 1.17)	0.71	0.80 (0.66, 0.96)	0.02
The COVID-19 pandemic is not under control in some countries after scaling up COVID-19 vaccination	N.A.	N.A.	0.98 (0.83, 1.17)	0.83
Infectiousness and harms of the variants of concern of SARS-CoV-2	N.A.	N.A.	0.83 (0.70, 0.99)	0.04
Outbreaks caused by variants of concern of SARS-CoV-2 in some places of China	N.A.	N.A.	1.07 (0.88, 1.29)	0.50
People develop COVID-19 after receiving a primary series of COVID-19 vaccination	N.A.	N.A.	1.24 (1.01, 1.52)	0.04

AOR: adjusted odds ratio, where the odds ratio was adjusted for significant background characteristics listed in Table 5. CI: confidence interval. N.A.: not applicable.

**Table 7 vaccines-10-01478-t007:** Associations between background characteristics and COVID-19 vaccine hesitancy for children aged 12–17 years.

	Round 1 (*n* = 254)		Round 2 (*n* = 340)	
	OR (95% CI)	*p*-Values	OR (95% CI)	*p*-Values
**Sociodemographic characteristics**				
Age, years	1.01 (0.95, 1.07)	0.87	0.83 (0.75, 0.92)	0.001
Gender				
Male	1.0		1.0	
Female	1.10 (0.58, 2.09)	0.77	1.08 (0.34, 3.48)	0.90
Relationship status				
Married	1.0		1.0	
Single or divorced	0.95 (0.29, 3.04)	0.93	7.07 (1.35, 37.03)	0.02
Having a stable partner	N.A.	N.A.	N.A.	N.A.
Education level				
Senior high or below	1.0		1.0	
College and above	0.70 (0.30, 1.61)	0.40	1.52 (0.48, 4.83)	0.47
Monthly personal income, CNY (USD)				
<3000 (462)	1.0		1.0	
3000–6999 (462–1077)	1.11 (0.57, 2.18)	0.76	0.27 (0.07, 1.06)	0.06
≥7000 (1078)	1.02 (0.32, 3.29)	0.97	0.37 (0.08, 1.71)	0.20
Type of work				
Frontline workers	1.0		1.0	
Management staff	0.95 (0.51, 1.77)	0.88	0.39 (0.08, 1.79)	0.22
**Personal COVID-19 preventive measures in the past month**				
Frequency of face mask wearing in public spaces or on transportation other than in the workplace				
Often/sometimes/never	1.0		1.0	
Every time	0.30 (0.10, 0.94)	0.04	0.16 (0.05, 0.52)	0.003
Frequency of face mask wearing when you have close contact with other people in the workplace				
Often/sometimes/never	1.0		1.0	
Every time	0.62 (0.31, 1.25)	0.18	1.53 (0.33, 7.15)	0.59
Frequency of sanitizing of hands by using soaps, liquid soaps, or alcohol-based sanitizer after returning from public spaces or touching public installations				
Often/sometimes/never	1.0		1.0	
Every time	0.92 (0.52, 1.62)	0.77	0.78 (0.25, 2.48)	0.68
Self-reported avoidance of social and meal gatherings with other people who do not live together				
No	1.0		1.0	
Yes	0.77 (0.44, 1.35)	0.36	0.97 (0.30, 3.11)	0.96
Self-reported avoidance of crowded places				
No	1.0		1.0	
Yes	0.93 (0.51, 1.71)	0.82	1.07 (0.31, 3.61)	0.92

OR: crude odds ratio. CI: confidence interval. N.A.: not applicable.

**Table 8 vaccines-10-01478-t008:** Factors associated with vaccine hesitancy for children aged 12–17 years.

	Round 1 (*n* = 254)		Round 2 (*n* = 340)	
	AOR (95% CI)	*p*-Values	AOR (95% CI)	*p*-Values
COVID-19 vaccine hesitancy among parents				
No	1.0			
Yes	1.59 (0.84, 3.01)	0.15	N.A.	N.A.
**Attitudes toward children’s COVID-19 vaccination**				
Positive Attitude Scale	0.71 (0.57, 0.88)	0.002	0.55 (0.34, 0.87)	0.01
Negative Attitude Scale	1.18 (0.99, 1.40)	0.07	1.25 (0.95, 1.66)	0.12
Perceived subjective norm related to child’s COVID-19 vaccination	0.33 (0.20, 0.54)	<0.001	0.26 (0.11, 0.61)	0.002
Perceived behavioral control to let the child be given a COVID-19 vaccination	0.51 (0.33, 0.80)	0.003	0.42 (0.18, 0.99)	0.049
**Frequency of exposure to the following information on social media (e.g., WeChat, WeChat moments, Weibo, Tiktok) in the past month**				
Experiences related to COVID-19 vaccination shared by recipients on social media	1.03 (0.79, 1.34)	0.85	0.62 (0.31, 1.26)	0.19
The COVID-19 pandemic is not under control in some countries after scaling up COVID-19 vaccination	N.A.	N.A.	0.59 (0.29, 1.22)	0.15
Infectiousness and harms of the variants of concern of SARS-CoV-2	N.A.	N.A.	0.76 (0.39, 1.48)	0.42
Outbreaks caused by variants of concern of SARS-CoV-2 in some places of China	N.A.	N.A.	0.74 (0.36, 1.49)	0.39
People develop COVID-19 after receiving a primary series of COVID-19 vaccination	N.A.	N.A.	0.75 (0.34, 1.69)	0.49

AOR: adjusted odds ratio, where the odds ratio was adjusted for significant background characteristics listed in Table 7. CI: confidence interval. N.A.: not applicable.

## Data Availability

The data presented in this study are available from the corresponding author upon request. The data are not publicly available as they contain personal behaviors.

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
