# Peer review of "Changes in Parents’ COVID-19 Vaccine Hesitancy for Children Aged 3–17 Years before and after the Rollout of the National Childhood COVID-19 Vaccination Program in China: Repeated Cross-Sectional Surveys"

_vaccines, 2022, doi:10.3390/vaccines10091478_

Round 1

Reviewer 1 Report

The authors report the results of two surveys conducted on factory workers from Shenzhen, China.  Although it is a well-written paper, the methodology used is not clear.

Methodological issues:

It is stated that this is a secondary analysis of a survey. It is not clear whether there was a primary study, if yes please provide the references of the study or preprint. Here also “Details of the recruitment and data collection were reported in published papers.”

Page 4 Line 145-149: 844 (first wave) and 1213 participants (second wave) who had at least one child aged 3-17 years were included in the analysis, while 2053 and 2626 of those eligible completed the surveys. If only eligible factory workers were approached and asked to participate (i.e., those who have children), this discrepancy in numbers is due to what?

What is the rationale behind changing the coding of vaccine hesitancy from “1=very unlikely, 2=unlikely, 3=neutral, 4=likely, and 5=very likely” to “very unlikely, unlikely or neutral to get vaccinated”? in those who are not yet vaccinated? This entire part is not very clear.

How was vaccine hesitancy defined in the two surveys given that answer coding was different?

It is not clear whether participants from the second survey are the same ones from the first survey. The statement in the discussion indicates so “factors at both individual and interpersonal levels”. If this is true, how come there are more participants in the second survey, if it is not how it was assured that participants do not repeat? Please declare clearly all inclusion and exclusion criteria and explain who the participants in the two surveys are.

How was assured that parents in the second survey were aware of the vaccination campaign?

Minor issues:

Please consider renaming the “wave” of the study in “study period”, or “survey period” or something else, since a wave can be confused with a pandemic wave, especially in Figure 1.

Page 2 line 53 “recommends that children aged five years or above to receive”, “should receive” not “to receive”.

Page 3 line 126: “Informed them” instead of “briefed them”.

Table 2: Please provide coefficients and ORs with 95%CI.

It is difficult to follow results for comparisons between the two surveys and two groups of parents in parallel, these parts of the results should be divided or reorganized.

It is not clear what “testimonials on social media” are, and whether it is a favorable or unfavorable behavior.

Please address explicitly the limitation that the results of the study are limited to workers who usually represent lower SES without high education.

Reviewer 2 Report

I was pleased to review the manuscript.

I have a few comments to improve its quality:

-       The main question is whether it was possible or not to ascertain if some parents answered both in the first and in the second wave or not

-       Was the response rate statistically different between the two surveys? If so, please discuss (try to explain why)

-       Recent data suggest that the experience of a previous SARS-CoV-2 infection modulates vaccine hesitancy (PMID: 35463893). If this was not considered, it should be discussed

-       The authors attribute a change due to the introduction of COVID-19 vaccine. However, other conditions they partially explored might explain the changes, e.g., the perception among parents that the new variants were more infectious among children. Also, the fear of restrictions for non-vaccinated children might have played a role. It is unclear to me if the authors investigated such aspects in both the surveys or only in the second one. Please clarify. If it was just investigated in the second one that should be acknowledged among the limitations

-        Please consider comparing the results of this study with previous surveys conducted in other settings/countries (e.g., PMID: 34696223, PMID: 35712618, PMID: 33999257). The current discussion is a bit too focused on China. I would make it more interesting also for a broader readership

-       Please also discuss vaccine hesitancy in parents also more in general (e.g. PMID: 33774718) and not strictly related to COVID-19

-       Please briefly report in the methods if the survey was validated or at least pilot tested

-       Survey/surveys should be uploaded as supplementary online material

-       in the current methods it is a bit difficult to clearly understand which questions were asked and in which wave. Might the authors include a new table to address this issue? This would help the readers

-  Tables are a bit difficult to read. Might the authors improve their layout?

Reviewer 3 Report

The longitudinal study on vaccine hesitancy is no doubt interesting and highly relevant for preventive health and public health preparedness. Nonetheless, I felt that these two concepts were not explicitly touch upon nor discussed at length. My suggestion would be to discuss about the notions of preventive health and public health preparedness as well as how the present longitudinal study extends understanding in these two areas.

A framework for preventive health marketing. Journal of Strategic Marketing.

The digital transformation of preventive telemedicine in France based on the use of connected wearable devices. Global Business and Organizational Excellence.

When it comes to the limitations, I do not think that the study should be stating cross-sectional design as a limitation when the study in fact is longitudinal. The title should also be amended to reflect this understanding.

There needs to be some explanation about future research directions.

One noteworthy concept that would make a good future contribution/direction is further investigation and scrutiny of behavioral control (covert, overt), which should help to provide targeted focus and strategies to deal with vaccine hesitancy.

Toward a theory of behavioral control. Journal of Strategic Marketing.

I hope these comments would be useful to help the authors improve their paper.

Good luck and all the very best.

Round 2

Reviewer 2 Report

The manuscript has markedly improved.  However a couple of pints have been addressed only in the rebuttal letter but should be considered also within the text:

-       The main question is whether it was possible or not to ascertain if some parents answered both in the first and in the second wave or not

-       Was the response rate statistically different between the two surveys? If so, please discuss (try to explain why)

After addressing these comments I will be happy to endorse the publication of this interesting article

Author Response

The manuscript has markedly improved. However a couple of pints have been addressed only in the rebuttal letter but should be considered also within the text:

  1. The main question is whether it was possible or not to ascertain if some parents answered both in the first and in the second wave or not
  2. Was the response rate statistically different between the two surveys? If so, please discuss (try to explain why)

After addressing these comments I will be happy to endorse the publication of this interesting article

A: Thank you for your comments. We have addressed these comments in the limitation section of the revised manuscript.  

“First, it was not possible to ascertain whether some parents answered both the first and second round of surveys as we did not ask participants in the second round whether they attended the first round.”

“The response rate was significantly higher in the second round as compared to the first round. The data collection method were identical between these two rounds of surveys. However, participants in the second round had a higher education level. It is possible that those with higher education would find it easier to complete an online survey and lead to a higher response rate. In our study, we controlled education level when comparing our outcome variables.”